# Improving Dispersion of Carbon Nanotubes in Natural Rubber by Using Waterjet-Produced Rubber Powder as a Carrier

**DOI:** 10.3390/polym15030477

**Published:** 2023-01-17

**Authors:** Xiurui Guo, Shouyun Guo, Gongxu Liu, Lichen Bai, Haichao Liu, Yuan Xu, Jinyang Zhao, Hailin Chai, Xingao Jian, Lei Guo, Fumin Liu

**Affiliations:** 1College of Electromechanical Engineering, Qingdao University of Science & Technology, Qingdao 266061, China; 2National Engineering Laboratory of Advanced Tire Equipment and Key Materials, Qingdao University of Science & Technology, Qingdao 266061, China; 3Sino-Thai International Rubber College, Qingdao University of Science & Technology, Qingdao 266061, China

**Keywords:** carbon nanotubes, dispersion, waterjet-produced rubber powder, properties of composites

## Abstract

Carbon nanotube (CNT), as reinforcing agents in natural rubber (NR), has gained a large amount of consideration due to their excellent properties. Uniform dispersion of CNT is the key to obtaining high-performance NR nanocomposites. In this contribution, a novel ultrasonic grinding dispersion method of CNT with waterjet-produced rubber powder (WPRP) as a carrier is proposed. Microscopic morphologies show that a Xanthium-like structure with WPRP as the core and CNTs as the spikes is formed, which significantly improves the dispersion of CNT in the NR matrix and simultaneously strengthens the bonding of the WPRP and NR matrix. With the increase in the WPRP loading, the Payne effect of CNT/WPRP/NR composites decreases, indicating the effectiveness of the dispersion method. The vulcanization MH and ML value and crosslinking density increase with the increase in the WPRP loading, whereas the scorch time and cure time exhibit a decreasing trend when the WPRP loading is less than 15 phr. It is found that the CNT/WPRP/NR composites filled with 5 phr WPRP have a 4% increase in 300% modulus, a 3% increase in tensile strength, while a 5% decrease in Akron abrasion loss, compared to CNT/NR composites.

## 1. Introduction

As an eminent reinforcing agent in natural rubber (NR), carbon nanotube (CNT) has gained a large amount of consideration in the rubber industry due to its excellent mechanical [1,2], thermal [3,4], and electrical [5,6] properties. The properties of CNT/NR composites mainly depend on the dispersion of CNT and the interfacial interaction between the rubber matrix and CNT. However, the dispersion of CNT is much more difficult than the other fillers on account of its large aspect ratio [7] and the large van der Waals force between CNTs [8,9]. Therefore, the dispersion method to obtain disaggregation and uniform distribution of CNT is the key to obtaining high-performance CNT/NR composites.

In previous work, four main dispersion methods have been reported in an attempt to prepare high-performance CNT/NR composites [10]: solution blending, latex blending, traditional mechanical blending, and surface modification of CNT. In the solution blending method, CNT is dispersed into a liquid rubber matrix by ultrasonication, stirring, or shear mixing [11]. Ismail [12] dissolved NR and CNT in two bottles of toluene, respectively. The two solutions were then mixed, and the mixture was dried before further processing. Although satisfactory dispersion of CNT in NR was obtained and the tensile properties and fatigue life of composites were enhanced using this method, solvent removal is a grave challenge [10]. Furthermore, the high cost of solvents and the environmental problems associated with solvents limits the industrial applications of the technology [13]. In the latex blending method, CNT is mixed with polymer latex, and the mixture was then dried [14]. Latex coagulation can also be adopted in this method by adding acid-treated CNT into negatively charged latex [15]. This method is more eco-friendly compared with solution blending. However, the large amount of deionized water used to disperse CNT dilutes the latex, which makes the CNT tend to settle in the diluted latex and leads to a secondary agglomeration [16]. Traditional mechanical blending disperses CNT in the matrix by strong shear mixing using an internal mixer, two-roll mill, ball mill, etc. [17,18]. Nakaramontri et al. [19] prepared CNT/ENR composites by mixing first in an internal mixer and then in a two-roll mill. However, CNT cannot be well dispersed in NR only by mechanical shear force [20]. Another method to obtain a uniform dispersion of CNT is surface modification. Abdul Lateef [21] introduced the carboxylic group on CNT’s surface by a nitric acid surface modification to improve the dispersion of CNT. The storage modulus of the modified composite was increased, which indicated a better matrix–filler interaction. However, one gram of the CNT should be modified with 10 mL of concentrated nitric acid at a temperature of 120 °C for 48 h and be washed with deionized water to remove excess nitric acid [21], which is a complicated process. Thus, it is imperative to propose an effective, low-cost, and easy method to obtain a uniform dispersion of CNT in NR.

Recently, Yihang Li [22] proposed a novel functional rubber vulcanization accelerator, petallike ZnO nanobundles grown on porous silica. It is reported that the agglomeration of nano-ZnO was suppressed because the ZnO was deposited in the silica pores. Using porous silica as the carrier for nano-ZnO, the curing time of the rubber was greatly reduced by 8.0−61.2% compared to using the filling made from mixed nano-ZnO and porous silica [22]. It can be deduced that the dispersion of CNT in NR can be improved by introducing appropriate intermediate carriers. As a kind of waste tire rubber production, waterjet-produced rubber powder (WPRP) prepared by high compressive shear stress and strong erosion from water jets process under entirely ambient conditions have irregular shapes with rough surfaces, large surface area, and strong surface activity [23]. Considering the surface area and the compatibility with the rubber matrix required for the carriers, the WPRP with a particle size of lower than 100 μm may be a promising intermediate carrier for CNT dispersion.

In the present study, a novel ultrasonic grinding dispersion method of CNT by using the WPRP with a particle size of lower than 100 μm as a carrier is proposed to achieve the dual goals of preparation of high-performance rubber composites and resource utilization of waste rubber. Various CNT/WPRP mixtures with different WPRP loading are provided to illustrate the effectiveness of the method. The transmission electron microscopy (TEM) of CNT/WPRP mixtures, the scanning electron microscopy (SEM) of CNT/WPRP/NR composites, as well as the “Payne effect” of the CNT/WPRP/NR composites, were investigated to analyze the dispersion of CNT. The vulcanization properties, tensile properties, Akron abrasion loss, and dynamic mechanical properties were measured to investigate the properties of CNT/WPRP/NR composites.

## 2. Materials and Methods

### 2.1. Materials

CNT (GT-400, diameter: 20–30 nm, length: 3–12 μm, specific surface area: 180–230 m^2^/g, Shandong Dazhan Nano Materials Co., Ltd., Zouping, China), WPRP (diameter ≤ 90 μm; specific surface area 5360 ± 10 m^2^/g; prepared from waste truck tire tread, which was vulcanized using 70 phr NR, 30 phr butadiene styrene rubber, 3 phr of zinc oxide (ZnO), 2 phr of stearic acid, 1 phr of paraffin wax, 1.5 phr of anti-aging agent 4010NA, 1 phr of anti-aging agent RD, 10 phr of oil, 2 phr of Gumalon, 43 phr of high abrasion furnace carbon black, 8 phr of Semi reinforced furnace carbon black, 0.65 phr of accelerator CZ, 0.15 phr pf accelerator DM, and 1.8 phr of sulfur; Dongguan Bingneng Rubber Co., Ltd., Dongguan, China), NR (consists of 96.89% poly(cis-l,4-isoprene) and a small percentage of biological elements, such as proteins, fatty acids and resins, and also inorganic materials; Hainan Natural Rubber Industry Group Co., Haikou, China), antioxidant 4020, stearic acid (SAD), ZnO, accelerator NS, sulfur, microcrystalline wax, antiscorcher CTP, and carbon black (CB N330; diameter about 22 nm; specific surface area 119 ± 5 m^2^/g; Shanghai Cabot Chemical Co., Ltd., Shanghai, China) are commercially available. Table 1 presents various formulas of the CNT/WPRP/NR composite adopted to investigate the effect of WPRP content, in which control formula 1, control formula 2, and control formula 3 are the formulas of original NR composites, CNT/NR composites and WPRP/NR composites, respectively. The phr of rubber ingredients in formula 2–6 is calculated according to the control formula 1 based on the rubber phr (including WPRP).

### 2.2. Preparation of CNT/WPRP/NR Composites

#### 2.2.1. Preparation of CNT/WPRP Mixtures

CNT was pre-dispersed in deionized water by ultrasonic dispersion instrument JH500W-20 (Hangzhou Jinghao Machinery Co., Ltd., Hangzhou, China) at 800 W and for 7 h under a 25 °C water bath, and then the CNT was forced dry at 40 °C. After drying, the CNT was ground and mixed with WPRP using ball mill 316 (Yongkang Electric Co., Ltd., Yongkang, China) for 5 min at 30,000 r/min and 3.5 kW. It should be noted that the grinder stopped for 10 min every 2.5 min to keep the material temperature constant. After the materials are mixed, they were left resting to be processed later.

#### 2.2.2. Preparation of CNT/WPRP/NR Composites

The CNT/WPRP/NR composites were prepared by an internal mixer XSM-500 (Shanghai Kechuang Rubber and Plastic Machinery Equipment Co., Ltd., Shanghai, China) at 60 r/min and 80 °C for 5 min, and other rubber ingredients were incorporated according to the formula given in Table 1. The CNT/WPRP composites and half of NR were added first into the internal mixer and mixed for 1 min, and then the other ingredients except sulfur and the remaining NR were added and mixed to the end of mixing time. The composites were then masticated in an open mill BL-6157 (Baolun Precision Testing Instrument Co., Ltd., Dongguan, China) at the roller spacing of 1.4 mm until the rubber compound wrapped the roller. After that, sulfur was added to the rubber compound at the roller spacing of 1.8 mm, punched with triangular wraps eight times, and rolled eight times. Finally, the roll distance of the open mill was adjusted to 0.8 mm for unloading. The plate vulcanizing machine QLB-400X400X2 (Shanghai No.1 Rubber Machinery Factory, Shanghai, China) was used to press the tablets (150 °C, 10 Mpa, *t*_90_ × 1.3 min). For comparison, original rubber without CNT, CNT/NR composites, WPRP/NR composite, as well as CNT/WPRP/NR composites with un-pre-dispersed CNT and WPRP according to formula 3 were also prepared by the same method.

### 2.3. Characterization and Measurements

#### 2.3.1. Characterization of the Dispersion of CNT and CNT/WPRP Mixtures

The dispersion of CNT in WPRP was analyzed by TEM using JEM2100 (JEOL Ltd., Tokyo, Japan) with an accelerating voltage of 200 kV. The dispersion of CNT/WPRP mixtures was analyzed by SEM using JSM7500F (JEOL Ltd., Tokyo, Japan); the sample was broken in liquid nitrogen, and the brittle section of the sample was vacuum gold coated before SEM imaging. “Payne effect” was analyzed by rubber process analyzer (RPA) using RPA2000 (Alpha Technologies Inc., Akron, OH, USA); the test configuration was a strain sweep mode in the range of 0.28–40%, the operating temperature was 60 °C, the operating frequency was 0.1 Hz.

#### 2.3.2. Characterization of the Properties of the CNT/WPRP/NR Composites

The vulcanization characteristics of the rubber compound were tested using a rotor-free rheometer MDR2000 (Alpha Technologies Inc., Akron, OH, USA) according to the standard ASTM D5289-2007a; the test temperature was 150 °C, and the maximum torque (MH), minimum torque (ML), scorch time (*t*_10_), as well as optimum curing time (*t*_90_) were measured. The tensile properties of the NR composites were measured using a tensile testing machine TS2005 (Taiwan Youken Technology Co., Ltd., Changhua, China) at a uniform crosshead speed of 500 mm/min according to the standard ASTM-D412. The hardness of the NR composites was tested using a shore durometer (LX-A, Shanghai Liuling Instrument Factory, Shanghai, China) based on the standard of ASTM-D2240. The Akron abrasion loss was measured using an Akron abrasion machine GT-7012-A (Taiwan GOTECH Technology Co., Ltd., Taizhong, China) according to the Chinese National Standards GB/T 1689–1998; test conditions were as follows: the testing temperature was 20 °C, the relative humidity was 8%, the number of revolutions was 3396, and the testing load was 500 g. The dynamic mechanical properties of the NR composites were performed by a dynamic mechanical analyzer (DMA) using GaBOME-TER-150 (Gabo Qualimeter, Ahlden, Germany) in tensile mode; the preset frequency was 1 Hz; the static strain and stress were 5% and 70 N, respectively; the dynamic strain and stress were 0.25% and 60 N, respectively; the scanning was conducted from −65 °C to 65 °C at a heating rate of 3 °C/min.

## 3. Results and Discussion

### 3.1. Ultrasonic Grinding Dispersion Mechanism of CNT

Figure 1 presents the ultrasonic grinding dispersion mechanism of CNT. It is reported that the tensile strength of CNT reaches more than 10 GPa and Young’s modulus varies from 270 to 950 GPa [24], which is much higher than the WPRP or rubber matrix. Therefore, the structure of individual CNT does not change during processing but can be implanted into the WPRP or rubber matrix under the action of mechanical force.

In the preparation of CNT/WPRP mixtures, ultrasonication is adopted to initially disentangle CNT. Due to the compatibility of the WPRP with the rubber matrix and its shape dimension suitability as a carrier for CNT, the dispersion of CNT is promoted in the grinding and mixing process of the CNT and WPRP compared to directly dispersing CNT in the matrix. Interaction between CNT and the WPRP makes the CNT not only wrap and adhere to the rough surface of the WPRP but also be implanted into the WPRP, forming a Xanthium-like three-dimensional structure with WPRP as the core and carbon nanotubes as the spikes. Therefore, the CNT attached to the WPRP carrier can be uniformly dispersed when blended with natural rubber, which improves the effect of CNT in reinforcing the rubber.

In the preparation of CNT/WPRP/NR composites, the rotation and migration of implanted Xanthium-like CNT/WPRP mixtures in the NR matrix under the action of mechanical force promote the dispersion of CNT wrapped and adhered on the surface of WPRP, which further improves the dispersion of fillers in the matrix. Meanwhile, the Xanthium-like CNT/WPRP mixtures entangle with macromolecular chains of NR due to the rotation and migration, forming plush ball-shaped structures with the Xanthium-like CNT/WPRP mixtures as the core, which enhances the bonding of the WPRP and NR matrix, makes the interface between the two materials indistinct, and improve the mechanical properties of the composites.

### 3.2. Dispersion of CNT

#### 3.2.1. Structural Characterization of CNT/WPRP Mixtures

TEM images presented in Figure 2 exhibited the dispersion of CNT in the CNT/WPRP mixtures to verify the dispersion mechanism of CNT. As shown in Figure 2a, the CNT was dispersed on the surface of the WPRP under the action of grinding. The structure of CNT wrapped on the WPRP, adhered to the surface of the WPRP, and implanted into the rubber powders can be observed in Areas 1, 2, and 3, respectively. This is in accordance with the mechanism proposed by us. Figure 2b presents the partial enlargement of Area 3 in Figure 2a. Since the CNT/WPRP mixtures cannot be penetrated by the electron beam [25], the CNT shown in Area 4 is only implanted into the surface of the WPRP. Although the intertwined CNT can still be observed, the interaction between WPRP and CNT promotes the dispersion of CNT under the action of grinding.

#### 3.2.2. Structural Characterization of CNT/WPRP/NR Composites

To verify the dispersion mechanism of CNT/WPRP mixtures and demonstrate the promoting effect of the Xanthium-like structure on the dispersion of CNT in natural rubber, the CNT/NR composites, CNT/WPRP/NR composites with un-pre-dispersed CNT and WPRP, as well as CNT/WPRP/NR composites with Xanthium-like CNT/WPRP mixtures prepared by the same blending method according to the formula 4 are compared in the microstructure. For comparison, the phr of CNT in all the samples mentioned above is 3. The SEM images are presented in Figure 3. Figure 3a presents the cross-sectional image of the CNT/NR composites at a magnification of 18,000 times. It can be observed that the untreated CNTs agglomerate with each other and separate from the rubber matrix, which is in accordance with the previous literature [8]. This is due to the strong van der Waals interactions between CNT, which can lead to stress concentration and thus reduce the mechanical properties of the composites. While in Figure 3b, almost no agglomerated CNT beam is found in the cross-sectional images of the CNT/WPRP/NR composites with Xanthium-like CNT/WPRP mixtures even at a magnification of 20,000 times, which indicates that the dispersion of CNT is significantly improved. Figure 3c presents the cross-sectional image of the CNT/WPRP/NR composites with un-pre-dispersed CNT and WPRP. It can be observed that there is an obvious interface between the WPRP and NR, indicating that the combination of the two is less effective. While in the cross-sectional images of the CNT/WPRP/NR composites with Xanthium-like CNT/WPRP mixtures shown in Figure 3d, the entire section is relatively flat, and there is no obvious interface between the WPRP and NR. This phenomenon verifies that the combination of the CNT-implanted WPRP and the NR matrix under the action of the mechanical force is improved, which is in accordance with the mechanism proposed by us.

#### 3.2.3. “Payne Effect”

To further illustrate the degree of agglomeration of the CNT in rubber composites, a “Payne effect” analysis is carried out. “Payne effect” refers to the phenomenon that the storage modulus (G′) of rubber composites decreased with the increase in strain [26]. In this contribution, “Payne effect” is quantified by the specific value ΔG′ (the difference between initial G′ 0.28% and final G′ 40%) [27]. Additionally, the smaller value of ΔG′, the better dispersion of the fillers. Figure 4 shows the RPA results of the composites with different formulas. Comparing the ΔG′ of the composites prepared with formula 1 and formula 2, it can be seen that the “Payne effect” of the rubber composite is significantly enhanced when CNT is added to the original rubber formula, which reflects the presence of weakly associating aggregates of CNT bundles formed because of the small size and large surface area of CNT [28]. By analyzing the ΔG′ of the composites with the formula 4–7, it can be found that the “Payne effect” of the CNT/WPRP/NR composites is weaker than that of the CNT/NR composites. When the filling amount of WPRP reaches 20 phr, ΔG′ reduced by 36.6% compared with that of CNT/NR composites. In addition the “Payne effect” continues to decrease with the increase in the loading amount of WPRP, which is contrary to the effect of the increase in rubber powder content on the “Payne effect” [29], suggesting that the ultrasonic grinding dispersion method of CNT by using WPRP as carrier enabled the CNT to be well dispersed into the rubber matrix.

### 3.3. Properties of the CNT/WPRP/NR Composites

Although the analysis in the previous section shows that the dispersion of CNT in the CNT/WPRP/NR composites prepared by the ultrasonic grinding dispersion method increases with the increase in WPRP content, the properties of the composites may be not improved as well, because the WPRP also affects composite properties. To obtain the appropriate WPRP content, the vulcanization properties, physical and mechanical properties, as well as dynamic mechanical analysis of the CNT/WPRP/NR composites are analyzed.

#### 3.3.1. Vulcanization Properties

Table 2 presents the vulcanization properties of the CNT/WPRP/NR composites. Comparing samples 1–3, it can be observed that the filling of both CNT and WPRP leads to an increase in the ML, MH, and the crosslinking density (difference between the MH and the ML), and the shortening of *t*_10_ and *t*_90_. The effect of WPRP is due to the presence of crosslinked precursors and unreacted curative in the rubber powder [30]. As for the CNT, the increase in the ML, MH and the crosslinking density is attributed to the formation of CNT-rubber matrix interactions [10]; and the high thermal conductivity in the axial direction of CNT can accelerate the vulcanization speed, leading to a shorten *t*_10_ and *t*_90_. Comparing samples 2 and 4–7, it can be observed that the ML, MH, and crosslinking density are increased with the increase in the WPRP content, which is related to the increase in WPRP content [31] and the improvement of CNT dispersion. The increase in WPRP content results in a better CNT dispersion effect, thereby the free movement of the NR molecular chains is restricted by the Xanthium-like CNT/WPRP mixtures, which further proves that the CNT/WPRP mixtures are well combined with NR and can promote rubber reinforcement. In addition, *t*_10_ and *t*_90_ are firstly decreased with the increase in the WPRP mixtures content. Additionally, when the filling amount of WPRP reaches 15 phr, *t*_10_ and *t*_90_ of the CNT/WPRP/NR composites are the lowest, which are reduced by 17.8% and 12.7%, respectively, compared with that of CNT/NR composites. This is because the well-dispersed Xanthium-like CNT/WPRP mixtures extend the good axial thermal conductivity of CNT to a three-dimensional space. The increase in *t*_10_ and *t*_90_ when the WPRP amount reaches 20 phr may be related to the decrease in thermal conductivity between rubber powder and rubber matrix due to excessive rubber powder. This indicates that the improvement of vulcanization properties of CNT/WPRP/NR composites is not only due to the increase in rubber powder content.

#### 3.3.2. Physical and Mechanical Properties

To confirm the improvement of rubber mechanical properties by the CNT/WPRP mixtures, the 300% modulus, elongation at break, tensile strength, hardness, and Akron abrasion loss of CNT/WPRP/NR composites were investigated, as shown in Figure 5. It can be observed from Figure 5a that although the 300% tensile modulus of the composites filled with WPRP only (sample 2) is decreased compared with that of the original NR composites, the 300% tensile modulus of all the CNT/WPRP/NR composites is increased. Among them, the modulus of the CNT/WPRP/NR composites filled with 5 phr WPRP is increased by 4% compared to the CNT/NR composites. This is because the crosslinking density is increased with the improvement of CNT dispersion. In addition, the 300% tensile modulus slightly decreases with increasing WPRP concentration, which is due to the weakening effect of WPRP [32].

Meanwhile, with the increase in the CNT/WPRP mixtures loading, the free space between rubber macromolecular chains becomes smaller, and the movement of chains is restricted, thereby reducing the elongation at the break of the rubber chain, just as shown in Figure 5b. In addition, it can be seen from Figure 5c that the tensile strength of the CNT/WPRP/NR composites decreases slightly with the increase in the WPRP content. This is because the large increase in WPRP loading leads to a decrease in tensile strength [30]. Additionally, the increase in the crosslinking density caused by the increase in WPRP content is mainly due to the formation of plush ball-shaped large rubber particles centered on the Xanthium-like CNT/WPRP mixtures. Under the action of large elongation, the disentanglement of rubber macromolecular chains leads to a decrease in tensile strength. Nevertheless, the tensile strength of CNT/WPRP/NR composites filled with 5 phr WPRP is increased by 3% compared to the CNT/NR composites, indicating the effectiveness of the dispersion method with WPRP as the carrier. To further play the role of CNT in enhancing mechanical properties, it is necessary to improve the dispersion state and interfacial bonding ability of CNT in the WPRP. This also provides a new direction for the recycling of waste rubber, which is worthy of further exploration.

Figure 5d,e presents the hardness and the Akron abrasion loss, respectively. It can be observed that the hardness and wear resistance of all CNT/WPRP/NR composites are improved compared with the original rubber. Among them, the CNT/WPRP/NR composites filled with 15 phr WPRP present the maximum hardness, which is increased by 9% compared to the original NR, while the CNT/WPRP/NR composites filled with 5–10 phr WPRP present the minimum Akron abrasion loss, which is decreased by 5% compared to the CNT/NR composites. With the increase in the CNT/WPRP mixtures loading, the hardness of the composites first increased and then decreased, while the trend of Akron abrasion loss is the opposite. This is because although the increase in CNT content can improve the cross-linking density in the network structure formed with the Xanthium-like CNT/WPRP mixture as the core, it will also adversely affect the hardness and wear resistance. When the content of WPRP is too high, the weakening effect will be greater than the strengthening effect caused by improving the dispersion of CNT.

#### 3.3.3. Dynamic Mechanical Analysis

To characterize the wet slip resistance and rolling resistance of the CNT/WPRP/NR composites, the tanδ and temperature curves measured by DMA are analyzed. Usually, the tanδ measured at 0 °C and low frequency is adopted to characterize the wet slip resistance of the composites, and the larger the tanδ, the better the wet slip resistance of the composites [33]. While the tanδ measured at 60 °C can be used to characterize the rolling resistance of the composites, and the smaller the tanδ, the lower the rolling resistance of the composites [34].

Table 3 presents the tanδ of the CNT/WPRP/NR composites at 0 °C and 60 °C. It can be observed that the wet slip resistance of the composites is very stable and higher than that of the original rubber. Additionally, comparing the tanδ at 60 °C of sample 1 and sample 2, it can be seen that the addition of CNT into the original rubber without WPRP as the carrier will lead to a sharp increase in the rolling resistance, which is due to the poor dispersion of CNT. Comparing samples 4–7, it can be observed that the rolling resistance decreases with the increase in the WPRP loading until it is similar to the original rubber. This is because the ultrasonic grinding dispersion method of CNT by using regenerated WPRP as a carrier improves the dispersion of CNT in the composites, which reduces the rolling resistance of the composites.

Figure 6 presents the relationship between the tanδ of the CNT/WPRP/NR composite and temperature. Compared with the original NR composites, the tanδ peak value of the CNT/WPRP/NR composite is skewed to the left, which means the glass transition temperature is decreased, indicating a better low-temperature resistance.

## 4. Conclusions

In the present study, the ultrasonic grinding dispersion method of CNT by using WPRP as a carrier was proposed to improve the dispersion of CNT in the NR matrix. Microscopic morphologies showed that a Xanthium-like structure with WPRP as the core and carbon nanotubes as the spikes was formed, which reduced the agglomeration of CNT and significantly improves the dispersion of CNT in the NR matrix. Meanwhile, the Xanthium-like CNT/WPRP mixtures entangled with macromolecular chains of NR during the mixing process and formed plush ball-shaped structures with the Xanthium-like CNT/WPRP mixtures as the core, which enhanced the bonding of the WPRP and NR matrix, made the interface between the two materials indistinct, and improved the mechanical properties of the composites. The “Payne effect” of the CNT/WPRP/NR composites was decreased with the increase in the WPRP loading, which further proved the effectiveness of the dispersion method with WPRP as a carrier.

Attributed to the improvement of CNT dispersion, the properties of the CNT/WPRP/NR composites were also improved. The vulcanization MH and ML value and crosslinking density were increased with the increase in the WPRP loading, whereas the scorch time and cure time exhibited a decreasing trend when the WPRP loading was less than 15 phr. Compared to the CNT/NR composites, the 300% modulus and tensile strength of the CNT/WPRP/NR composites filled with 5 phr WPRP were increased by 4% and 3%, respectively, and the Akron abrasion loss was decreased by 5%. In addition, the CNT/WPRP/NR composites prepared also showed better wet slip resistance, lower rolling resistance and better low-temperature resistance compared with the original NR composites. The contribution not only provides an effective method for dispersing CNT in NR but also provides a new idea for the recycling of waste rubber, which may play a significant role in the rubber industry.

## Figures and Tables

**Figure 1 polymers-15-00477-f001:**
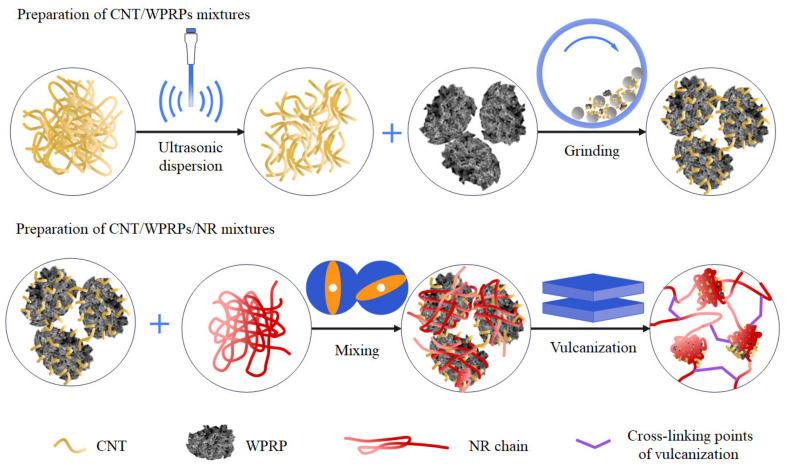
Ultrasonic grinding dispersion mechanism of CNT.

**Figure 2 polymers-15-00477-f002:**
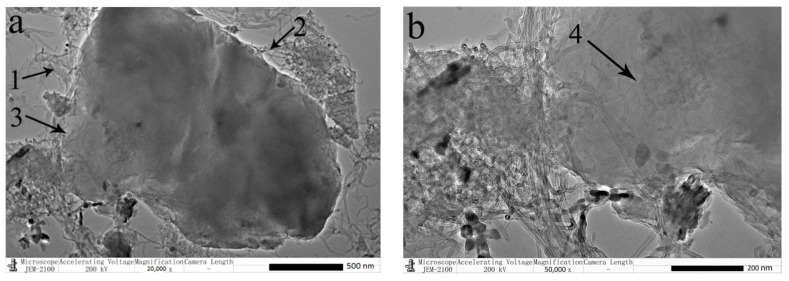
TEM images of CNT/WPRP mixtures (**a**) ×20,000 (**b**) ×50,000.

**Figure 3 polymers-15-00477-f003:**
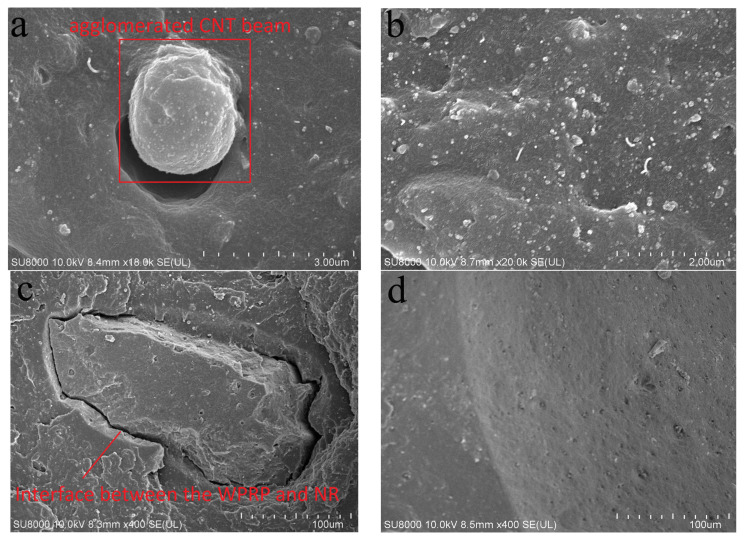
SEM images of (**a**) CNT/NR composites; (**b**) CNT/WPRP/NR composites with Xanthium-like CNT/WPRP mixtures (×20,000); (**c**) CNT/WPRP/NR composites with un-pre-dispersed CNT and WPRP; and (**d**) CNT/WPRP/NR composites with Xanthium-like CNT/WPRP mixtures (×400).

**Figure 4 polymers-15-00477-f004:**
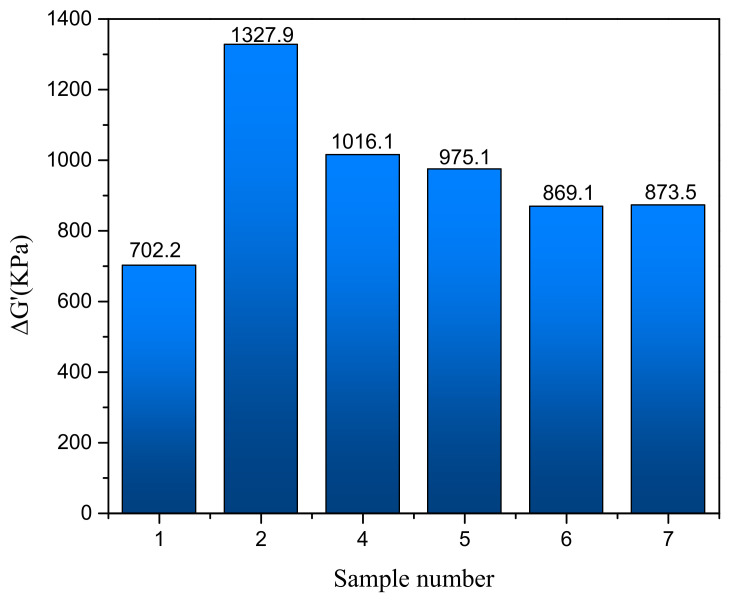
RPA results of the composites.

**Figure 5 polymers-15-00477-f005:**
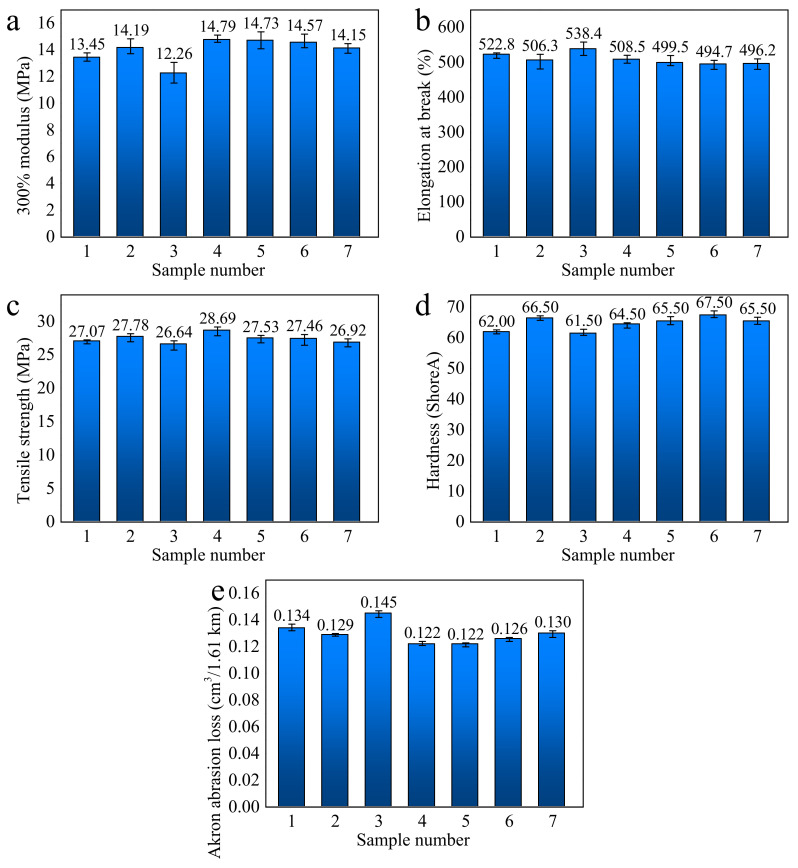
Physical and mechanical properties of the composites including (**a**) 300% modulus, (**b**) elongation at break, (**c**) tensile strength, (**d**) hardness, and (**e**) Akron abrasion loss.

**Figure 6 polymers-15-00477-f006:**
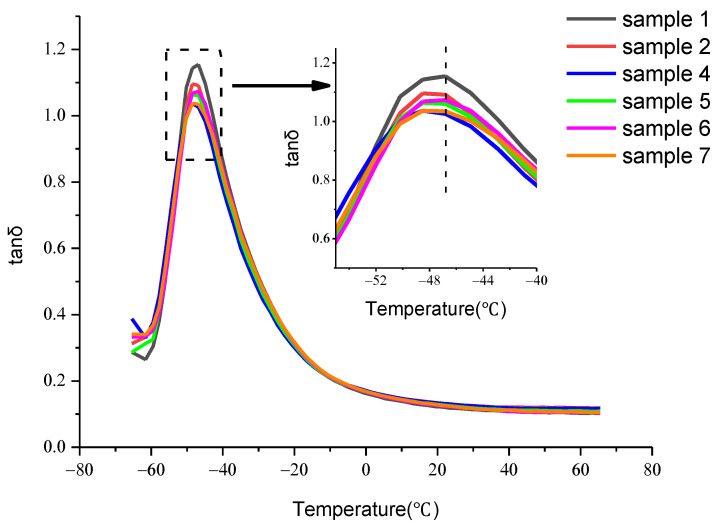
Relationship between the tanδ of the CNT/WPRP/NR composite and temperature.

**Table 1 polymers-15-00477-t001:** The formulas of the CNT/WPRP/NR composite.

Ingredient	phr						
	1	2	3	4	5	6	7
NR	100	100	100	100	100	100	100
ZnO	3	3	3.3	3.15	3.3	3.45	3.6
Stearic acid	1	1	1.1	1.05	1.1	1.15	1.2
NS	1.6	1.6	1.76	1.68	1.76	1.84	1.92
Sulfur	1.2	1.2	1.32	1.26	1.32	1.38	1.44
Antioxidant (4020)	2	2	2.2	2.1	2.2	2.3	2.4
Microcrystalline wax	1	1	1.1	1.05	1.1	1.15	1.2
CTP	0.3	0.3	0.33	0.315	0.33	0.345	0.36
CB (N234)	50	50	50	50	50	50	50
Mixture of CNT/WPRP	0/0	3/0	0/10	3/5	3/10	3/15	3/20

**Table 2 polymers-15-00477-t002:** Vulcanization characteristics of the CNT/WPRP/NR composites.

Sample Number	MH nm	ML nm	MH-ML nm	*t*_10_ min	*t*_90_ min
1	17.65	1.59	16.06	3.94	7.98
2	19.40	2.16	17.24	3.03	6.91
3	18.14	1.92	16.22	3.54	7.35
4	19.58	2.34	17.24	2.82	7.03
5	19.62	2.35	17.27	2.59	6.85
6	19.75	2.49	17.26	2.49	6.03
7	20.45	2.63	17.82	2.63	6.26

**Table 3 polymers-15-00477-t003:** Tanδ of the CNT/WPRP/NR composites at 0 °C and 60 °C.

Sample Number	1	2	4	5	6	7
Tanδ (0 °C)	0.163	0.173	0.170	0.170	0.168	0.170
Tanδ (60 °C)	0.102	0.119	0.117	0.110	0.106	0.105

## Data Availability

Not applicable.

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
