# Peer review of "Improving Dispersion of Carbon Nanotubes in Natural Rubber by Using Waterjet-Produced Rubber Powder as a Carrier"

_polymers, 2023, doi:10.3390/polym15030477_

Round 1
Reviewer 1 Report
The authors explained the use of WPRP to assist CNT dispersion in NR composites. They showed that the overall properties of the composites were improved comparing with untreated CNT. The study was well conducted. However, there are two points to be considered.
1) The authors explained the changes properties by crosslink density. Thus, the characterization for the crosslink density should be included.
2) More references in the Results & Discussion part should be included.
Reviewer 2 Report
The subject of the paper is interesting, but the text must be improved before being accepted. The main issues are:
- The English should be improved, especially concerning the right use of articles.
- Please remove the instructions to authors in the beginning of the introduction.
- Please define NR, CNT and WPRP in the introduction, not only in the abstract.
- On page 2, line 66, it says "Storage of the modified composite was 66 increased, which indicated a better matrix-filler interaction". I suppose the authors mean "storage modulus".
- The main point that requires more clarity is explaining how the CNT/WPRP mixture was prepared. How was CNT initially dispersed by ultrasound? Was it in water? And please describe the type of grinder used to mix it with WPRP. If water was added, how was it removed afterwards?
- What do the autors mean by “Since the size of the WPRP is much smaller than that of the rubber matrix,”? (Page 5, line 165)
- Please improve the scale bar in Figure 2.
Reviewer 3 Report
The manuscript's topic is interesting. However, the work requires considerable correction. And additional explanations are necessary. After the author's answer, it is necessary to do a detailed review once again.
I ask the authors to make the following corrections and respond to the following comments:
Introduction Delete the first three sentences in the introductory part. The work is started with the instructions for the preparation of the work! These are the sentences: “As far as possible, please keep the introduction comprehensible to scientists outside your particular field of research. References should be numbered in order of appearance and indicated by a numeral or numerals in square brackets—e.g., [1] or [2,3], or [4–6]. See the end of the document for further details on references.”
It is necessary to cite in which paper? Lines 45-47: “In previous work, four mainly dispersion methods have been reported in an attempt to prepare high-performance CNT/NR composites: solution blending, latex blending, traditional mechanical blending, and surface modification of CNT.”
In the introductory part, the comparison effect of waste tire rubber production with the effect of an accelerator is inadequate. In the introductory part, the comparison effect of waste tire rubber production with the impact of an accelerator is inadequate. The role of nano-ZnO in these systems is different. In the literature, many papers in which waste tire rubber production in new composite based on the rubber is added, and the conclusions in those papers should be presented. “Recently, Yihang Li [22] proposed a novel functional rubber vulcanization accelerator, petallike ZnO nanobundles grown on porous silica. It is reported that the agglomeration of nano-ZnO was suppressed because the ZnO was deposited in the silica pores. Using porous silica as the carrier for nano-ZnO, the curing time of the rubber was greatly reduced by 8.0−61.2 % compared to using the filling made from mixed nano-ZnO and porous silica [22].”
WPRP needs to be characterized. Provide basic information. What are the particle size and specific surface area? What is the composition? We assume there are different nonrubber components? (ZnO, sulfur, and others) Their very presence can be crucial. It is necessary to characterize NR. Provide primary data important for NR.
It is necessary to provide essential data for carbon black. What is the particle size?
Preparation of CNT/WPRP/NR composites needs to be described in more detail. For example, in Lines 118-119: “and other rubber ingredients were incorporated according to the formula.” What formula? How were the ingredients added? In what time interval?
Lines 119-120: “Then the composites were blended with sulfur in an open mill BL-6157” Describe the conditions. How long?
Line 121-122: How long are the samples pressed?
References to the literature need to be included in the interpretation of the results, and it needs to be clarified on what basis some conclusions are drawn. For example, on what basis was this concluded: “Interaction between CNT and the WPRP makes the CNT not only wrap and adhere to the rough surface of the WPRP but also be implanted into the WPRP, forming a Xanthium-like three-dimensional structure with WPRP as the core and carbon nanotubes as the spikes.”
Are you sure the SEM images prove that the ingredient is well dispersed? And that you have shown the areas in the material where they are? It is necessary to mark on the SEM image where CNT/VPRP is observed in the NR matrix and to mark that good compatibility has been achieved.
Round 2
Reviewer 1 Report
The manuscript can be accepted in its current form.
Reviewer 2 Report
The paper can now be accepted.
Reviewer 3 Report
I suggest that the revised manuscript be accepted.